# High Expression Level of PPARγ in CD24 Knockout Mice and Gender-Specific Metabolic Changes: A Model of Insulin-Sensitive Obesity

**DOI:** 10.3390/jpm11010050

**Published:** 2021-01-15

**Authors:** Shiran Shapira, Dina Kazanov, Rachel Dankner, Sigal Fishman, Naftali Stern, Nadir Arber

**Affiliations:** 1The Integrated Cancer Prevention Center, Tel Aviv Sourasky Medical Center, Tel Aviv 6423906, Israel; shiranshapira@gmail.com (S.S.); dianak@tlvmc.gov.il (D.K.); 2Department of Molecular Genetics and Biochemistry, Sackler Faculty of Medicine, Tel Aviv University, Tel Aviv 69978, Israel; 3Unit for Cardiovascular Epidemiology, The Gertner Institute for Epidemiology and Health Policy Research, Sheba Medical Center, Tel Hashomer 52621, Israel; racheld@gertner.health.gov.il; 4Department of Epidemiology and Preventive Medicine, Sackler Faculty of Medicine, School of Public Health, Tel Aviv University, Ramat Aviv, Tel Aviv 69978, Israel; 5Bariatric Endoscopy Service, Department of Gastroenterology and Liver Disease, Tel Aviv Sourasky Medical Center, affiliated with the Sackler School of Medicine, Tel Aviv University, Tel Aviv 69978, Israel; sigalf@tlvmc.gov.il; 6The Sagol Center for Epigenetics of Aging and Metabolism, Institute of Endocrinology, Metabolism and Hypertension, Tel Aviv-Sourasky Medical Center and Sackler Faculty of Medicine, Tel Aviv University, Tel Aviv 69978, Israel; nafstern66@gmail.com

**Keywords:** PPARγ, CD24, insulin sensitivity, diabetes, cancer

## Abstract

Background: The heat-stable HSA/CD24 gene encodes a protein that shows high expression levels in adipocyte precursor cells but low levels in terminally differentiated adipocytes. Its high expression in many types of human cancer suggests an association between cancer, diabetes, and obesity, which is currently unclear. In addition, peroxisome proliferator-activated receptor gamma (PPARγ) is a regulator of adipogenesis that plays a role in insulin sensitivity, lipid metabolism, and adipokine expression in adipocytes. Aim: To assess gender-dependent changes in CD24 KO and its association with PPARγ expression. Experimental approach: WT and CD24 KO mice were monitored from birth up to 12 months, and various physiological and molecular characteristics were analysed. Mean body weight and adipose mass were higher in KO mice than in WT mice. Male, but not female, KO mice showed increased insulin sensitivity, glucose uptake, adipocyte size, and PPARγ expression than WT mice. In addition, enteric bacterial populations, assessed through high-throughput sequencing of stool 16S rRNA genes, were significantly different between male KO and WT mice. Conclusions: CD24 may negatively regulate PPARγ expression in male mice. Furthermore, the association between the CD24 and insulin sensitivity suggests a possible mechanism for diabetes as a cancer risk factor. Finally, CD24 KO male mice may serve as a model of obesity and insulin hyper-sensitivity.

## 1. Introduction

The heat stable antigen (HSA, or murine CD24) is a heavily glycosylated glycosyl phosphatidylinositol-anchored protein expressed on most immature hematopoietic lineages, developing neurons and epithelial cells. Its expression is dynamically regulated during cellular differentiation, whereas its expression is usually absent from cells that have reached their final differentiation [1].

The role of CD24 in carcinogenesis is increasingly becoming a focus of interest [2,3,4,5,6]. CD24 is highly expressed in a large variety of human cancers and is involved in processes such as proliferation, invasion, chemosensitivity, and metastasis of various types of cancer [7]. In addition, CD24 has been associated with diabetes, possibly through the associations between cancer, diabetes, and obesity [8,9,10,11,12], for which several molecular mechanisms have been proposed, including the insulin synthesis pathway and glucose regulation. In the synthesis of insulin, C-peptide is cleaved from proinsulin to form mature insulin. Increased insulin secretion may lead to increased hepatic growth hormone-mediated synthesis of IGF-1. High levels of insulin, C-peptide, and IGF-1 have been associated with an increased risk of certain cancers in epidemiological studies, including breast cancer, colorectal cancer, and prostate cancer. Regarding glucose homeostasis, glucose transporters GLUT1, GLUT3, and GLUT4 are known to be overexpressed in many tumors. Furthermore, the upregulation of anaerobic glycolysis directs glycolytic intermediates into the pentose phosphate pathway, leading to the production of precursors for fatty acids, amino acids, and nucleic acids. In hyperglycemic mice, more aggressive skin and mammary tumors were found compared to normoglycemic mice. Additional pathways connecting diabetes, obesity, and cancer include estrogen signaling, dyslipidemia, and inflammatory pathways [7]. In addition to these examples, CD24 is known to support the differentiation of immature pre-adipocytes into adipocytes [13,14,15].

However, the involvement of CD24 in diabetes is not entirely clear. On the one hand, daughter cells of CD24-negative pancreatic ductal cells have been shown to differentiate into insulin-secreting cells in vitro [16]. On the other hand, CD24-positive human embryonic stem cells (hESCs), but not CD24-negative cells, have been reported to be capable of differentiating into insulin-secreting cells, although these findings were later questioned in a study of both mouse and human ESCs [17]. Furthermore, in Hashimoto’s thyroiditis (HT), associated with insulin resistance, the level of CD19^+^ CD24^hi^CD38^hi^ regulatory B-cells (Bregs) was reduced, suggesting a potential role for CD24 in insulin resistance in these patients [18]. A recent full single-cell transcriptome of pancreatic cells showed that CD24 and TM4SF4 expression can be used to sort live alpha and beta cells with high purity [19].

Insulin resistance and obesity are intricately related, and both are considered as causative factors in the development of type 2 diabetes (T2D) [18,19]. The association between obesity and impaired insulin-sensitivity has long been recognized, although some obese individuals seem to be protected from insulin resistance [20].

Referred to as the ultimate thrifty gene [21], peroxisome proliferator-activated receptor gamma (PPARγ) has been identified as the master regulator of adipogenesis. PPARγ plays a role in insulin sensitivity, lipid metabolism, and adipokine expression in adipocytes. It confers an advantage when food supply is unstable but increases the risk for obesity and diabetes when food is abundant. The activation of PPARγ, which occurs through ligand binding, results in a marked improvement of insulin and glucose parameters in T2D patients, resulting from an increase of whole-body insulin sensitivity [22,23].

In humans, there are two general approaches when evaluating differences in substrate metabolism between males and females. One, which is clinically probably the most relevant, is to accept the differences in phenotype between men and women and acknowledge that the observed differences in metabolism may be secondary to those characteristics. The other strives to eliminate as many as practically feasible potentially confounding variables to determine if gender per se (i.e., sexual genotype) affects the control of metabolism. It is thus of importance to consider gender differences when investigating the associations of CD24 with obesity and weight gain. In addition, the influence of sex on the development of obesity, Type 2 Diabetes Mellitus (T2DM), and dyslipidemia is well documented, although the molecular mechanism underlying those differences reminds elusive [24,25,26,27,28]. In addition, ligands of peroxisome proliferator-activated receptors (PPARs) are used as oral antidiabetics (PPARγ agonists: thiazolidinediones, TZDs), or for the treatment of dyslipidemia and cardiovascular diseases, due to their lipid-lowering properties (PPARα agonists: fibrates), as PPARs control transcription of a set of genes involved in the regulation of lipid and carbohydrate metabolism. Therefore, investigations of the association between CD24 and obesity and weight gain in humans should consider gender-dependent associations of PPARγ.

In 1997, the generation, production and characterization of mice that completely lack HSA expression was reported. CD24 knockout mice exhibit a loss of immature B cells via their increased apoptosis demonstrating that altering CD24 expression has serious repercussions in these cells [29,30]. In addition, male CD24 KO mice show a significant decrease in white adipose tissue of more than 40%, as well as increased fasting glucose and free fatty acids, decreased fasting insulin, and plasma leptin. It was concluded that the global absence of CD24 affects adipocyte cell size and causes metabolic disturbances in glucose homeostasis and free fatty acid levels [14]. In this study, we aimed to assess gender-dependent changes in CD24 KO and its association with PPARγ expression.

## 2. Materials and Methods

### 2.1. Materials

All reagents were purchased from Sigma-Aldrich (Rehovot, Israel), unless otherwise stated. EZ-ECL detection kit and cell culture supplements were purchased from Beit-Haemek, Israel.

### 2.2. Transgenic Mice

Mice of genotypes CD24^−/−^ (KO) and CD24^+/+^ (WT) were bred on the same genetic background (C57B/6J × C57BL/6). The mice were bred in our facility and were 8–12 weeks-old at the start of the experiment. All had ad libitum access to a standard pellet diet and to tap water and were kept in an animal room at constant temperature (22 ± 2 °C) with a standard 12 h light/dark cycle. Food was weighed twice a week. Body weight and water and food consumption were closely monitored from birth and compared between KO and WT mice for 12 months. All experiments were approved by the institutional committee for animal welfare at Tel Aviv Sourasky Medical Center.

In the case of high fat diet consumption, a rodent diet with 60 kcal% fat was used (Research Diets, D12492, see Table 1. Mice that were fed with HFD received this diet from the age of 3–4 weeks, at the end of breastfeeding.

### 2.3. Genotyping of CD24 Knockout Mice

Genomic DNA was extracted from mouse tails, and the genotype of HSA^−/−^ was verified by Polymerase chain reaction (PCR). Two PCRs were conducted to detect the CD24 Exon 1 fragment and/or the Neomycin cassette, which replaces the CD24 Exon1 in two alleles of CD24^−/−^ genotype (and in one allele of CD24^+/−^). Table 2 summarizes the oligonucleotides and the reaction details used for this assay. The reaction was carried out in a Tgradient ThermoCycler (Biometra, Germany) using 2x ReddyMix™ PCR Master Mix (Thermo Scientific, Rhenium Israel, cat. No. AB-0575/DC/LD/B).

### 2.4. Genotype Verification by Flow Cytometry Analysis

Heparinized peripheral blood samples (100 µL) were collected from the orbital sinus of KO and WT (control mice strain C57B/6J) and diluted in PBS. The samples were then centrifuged at 2000 rpm for 5 min. The pellet was incubated with 100 µL of rat anti-CD24 antibody, clone M1/69 at 4 °C for 30 min. Cells were washed twice in FACS buffer (10% FCS, 0.01% sodium azide in ice cold PBS). Then, 100 µL of 1:100 diluted fluorescein isothiocyanate (FITC)-conjugated goat anti-rat antibody (Jackson Immunoresearch Laboratories, Inc, West Grove, PA, USA) was added to the pellet and incubated at 4 °C for 30 min. Cells were washed twice with FACS buffer and bound antibodies were detected on a FACSCalibur™ and analyzed using CellQuestion software (both supplied by Becton Dickinson, San Jose, CA, USA).

### 2.5. Insulin Sensitivity

Insulin sensitivity in KO and WT mice (*n* = 5–10 per genotype) was determined at 9–12 weeks of age through measurement of blood glucose reduction following an insulin tolerance test. Both groups were fasted overnight (12 h) and then intraperitoneally (IP) injected with 0.5 units insulin per kg of body weight. Whole-blood glucose was determined at 0, 15, 30, 45, 90, and 120 min following the insulin injection, and the data were presented as a percentage of baseline glucose levels at T = 0. Tail-tip whole-blood glucose was determined using an Accutrend^®^ Sensor (Roche Diagnostics, Basel, Switzerland), using blood glucose test strips as described by the manufacturer.

### 2.6. Intraperitoneal Glucose Challenge

The blood glucose response to IP glucose administration in KO and WT mice (*n* = 5–10 per genotype) was determined at 9–12 weeks of age through measurement of blood glucose elevation following a glucose challenge test. Both groups were fasted overnight (12 h) and then injected IP with 1 g glucose (Sigma, Israel, cat. no. 47829) per kg of body weight. Whole-blood glucose was determined at 0, 20, 40, 60, 90, and 135 min following the glucose injection, and the data were presented as a percentage of the glucose level at T = 0. Tail-tip whole-blood glucose was determined with the Accutrend^®^ Sensor (Roche Diagnostics, Basel, Switzerland), using blood glucose test strips as described by the manufacturer.

### 2.7. Triglycerides in Liver Tissue

Approximately 200 mg of frozen liver tissue, isolated from KO and WT mice, was homogenized in 4 mL 2:1 chloroform (Sigma, Rehovot, Israel, cat. no. C-2432): methanol (Biolab, Beit Haemek, Israel, cat. no. 001368052100) solution, lightly vortexed and incubated for 10 min at room temperature (RT). Samples were filtered through Sharkskin filter (Fisher Scientific, Waltham, MA, USA) and 0.2 volumes of 0.58% NaCl solution were added. Samples were centrifuged at 4 °C at 1000 rpm for 5 min and the upper phase was gently aspirated. Samples were then washed again with NaCl and stored until the analysis was performed.

Samples were evaporated by gently blowing with nitrogen, reconstituted by resuspension in 1 mL of Triton™ X-100 (Sigma, Israel, cat. no. T8787) (5% *v*/*v* in water), and slowly heated in a water bath to 80 °C. Finally, samples were gradually cooled to RT and then heated again for triglyceride solubilization. Insoluble residues were removed by centrifugation at 4 °C for 5 min at 1000 rpm and the supernatant was diluted 10-fold to a final volume of 10 mL with distilled water for analysis.

### 2.8. Total RNA Extraction from Animal Tissues

Kidney adipose tissues were isolated from KO and WT mice and total RNA was extracted as follows: the tissue was homogenized in 4.5 mL TRI reagent^®^ (Sigma Aldrich, Rehovot, Israel, T9424) and incubated at RT for 5 min. Samples were divided on ice into three samples and 0.3 mL of chloroform was added to each sample. After gentle mixing, samples were incubated at RT for 8 min and then centrifuged at 4 °C for 15 min at 13,000 rpm. The upper phase was transferred to another tube and 0.75 mL of isopropanol was added. After gentle mixing, samples were incubated at RT for 7 min and then centrifuged at 4 °C for 15 min at 13,000 rpm. The pellet was washed twice with 75% ethanol. The resulting RNA pellet in the three tubes was then recombined, resuspended in double distilled water, and heated to 55 °C for 2–5 min. Pure RNA concentration was determined using a NanoDrop^TM^2000 device (Thermo Scientific, Waltham, MA, USA).

### 2.9. cDNA Synthesis and PCR Amplification of PPARγ

For RT-PCR reactions, 3–5 µg of pure RNA was used. cDNA synthesis was performed according to the manufacturer′s instructions (Promega, WI, USA). The RT-PCR products were amplified in a PCR reaction using the following primer sequences targeted against the PPARγ gene: forward: 5′-GCGGAGATCTCCAGTGATATC-3′, reverse: 5′-CACCAAAGGGCTTCCGC-3′. The mouse GAPDH gene was used as a reference housekeeping gene and amplified using the following primer sequences: forward: 5′-GGAGATTGTTGCCATCAACG-3′, reverse: 5′-TTGGTGGTGCAGGATGCATT-3′. The amplified DNA products were identified by electrophoresis using 2% agarose gels and using GelStar™ Nucleic Acid Gel Stain (Lonza, Basel, Switzerland). They were then visualized under UV light.

### 2.10. Adipocyte Cultures

For the isolation and culture of primary adipose cells, 15-week-old male KO and WT mice were used to obtain primary adipose cells essentially as described previously [31]. Briefly, the epididymal fat pads were removed, minced, and digested using collagenase at 37 °C for 2 h. The primary adipose cells were then washed extensively and incubated at 37 °C in a KRBH buffer (Krebs-Ringer-bicarbonate HEPES buffer, pH 7.4) or Dulbecco’s modified Eagle’s medium containing 5% bovine serum albumin. Primary adipose cells and conditioned medium were taken at various times as indicated in the figure legends and were flash-frozen in liquid nitrogen and stored in 80 °C until use.

### 2.11. qRT-PCR for PPARγ Expression in Visceral Fat

For each reaction, 0.02 µg/mL of pure RNA was used. A StepOne™ Real-Time PCR System (Applied Biosystems, USA) was used according to the manufacturer’s instructions with a SYBR^®^ Select Master Mix (Thermo Fisher, USA) with SYBR^®^ GreenER™ dye for visualization. The RT-PCR products were amplified in a PCR reaction using the following primer sequences targeted against the PPARγ gene: forward: 5′-CATAAAGTCCTTCCCGCTGA-3′, reverse: 5′-GAAACTGGCACCCTTGAAAA-3′. The mouse RPLPO gene was used as a reference housekeeping gene and amplified using the following primer sequences: forward: 5′-TCCAGCAGGTGTTTGACAAC-3′, reverse: 5′-CCGATCTGCAGACACACACT-3′. The results were analyzed using the StepOne™ Software according to the manufacturer’s instructions (Applied Biosystems, Forster City, CA, USA).

### 2.12. Stool 16S rRNA Genes Assessment

Fecal samples were collected from WT and KO mice (*n* = 8–11 for each genotype) receiving either a regular (NC) or a high-fat diet (HFD). All samples were placed immediately into sterile plastic tubes and stored at −80 °C until analysis. DNA was extracted from the samples using the FastDNA SPIN Kit for Soil (MP Biomedicals Inc., Solon, OH, USA) according to the manufacturer’s instructions.

PCR amplification was performed using primers targeting from V3 to V4 regions of the 16S rRNA gene with extracted DNA. The PCR conditions used were 5 min at 95 °C, 35 cycles of 30 s at 94 °C, 30 s at 55 °C and 90 s at 72 °C, followed by 10 min at 72 °C. Amplification was carried out by using a Verity Thermocycler (Applied Biosystems, Forster City, CA, USA). The PCR product was confirmed by using 2% agarose gel electrophoresis and visualized under UV light. The amplified products were purified with the Wizard SV Gen PCR Clean-Up System (Promega, WI, USA). Equal concentrations of purified products were pooled together and followed by a further purification step involving the Agencourt AMPure XP DNA purification beads (Beckman Coulter Genomics GmbH, Bernried, Germany) in order to remove primer dimers. The quality and product size were assessed using a DNA 7500 chip. Mixed amplicons were pooled and the sequencing was carried out according to the manufacturer’s instructions.

### 2.13. Statistical Analysis

All experiments were performed at least in triplicates. Quantitative data were analyzed using the student *t*-test. For all tests, *p* < 0.05 was considered significant. Data were expressed as the mean ± standard deviation (SD).

The operational taxonomic unit (OTU) table of raw counts was normalized to an OTU table of relative abundance values. Taxa analysis was performed on the core taxa prevalent in more than >25% of samples. Same types of taxa were agglomerated at the phylum, class, order, family, and genus level. We used unweighted and weighted Unifrac distance of even OTU samples to perform Principal Coordinate Analyses (PCoA) and Analysis of similarities (ANOSIM) was used to analyze the difference among groups. Linear discriminant analysis (LDA) Effect Size (LEfSe) was performed to find out the differentially enriched taxa between groups. The functional prediction of microbiota was done with PICRUSt [32]. Only reads identified in closed reference picking (Greengenes 13_5 database) were used for the PICRUSt analysis. The reference genome coverage of samples was also calculated using weighted Nearest Sequenced Taxon Index (NSTI) score with the -a option in the predict metagenomes.py script.

## 3. Results

### 3.1. Establishment of CD24 Knockout Mice Colony

A colony of KO mice was generated by breeding and crossing them over. The genotype of the mice was verified by FACS analysis, based on the absence of CD24 expression on the surface of erythrocytes from the KO mice. The genetically altered mice were demonstrated to be completely inbred: the erythrocytes of the WT were stained (Figure 1A), while those of the KO mice were not (Figure 1B). The KO genotype was confirmed by Western blot analysis (Figure 1C) using specific anti-CD24 M1/69 monoclonal antibodies. A similar colony of WT C57B/6J mice served as a control group.

### 3.2. CD24 Deficiency Is Associated with Increased Body Weight and Increased Abdominal Fat Mass in a Gender-Dependent Manner

The phenotype and possible developmental changes of the mice were closely monitored, including monitoring of food consumption, for 12 months. The mean body weight, weighed weekly, of the KO male mice was greater than that of WT male mice of the same body length and age (Figure 2A,B). The weight difference was noted from week 4 and lasted until 12 months, when the study was completed. These differences were not apparent in female mice. Water and food consumption were similar in KO and WT mice over the course of a year (data not shown). Figure 2C shows the weight follow-up of male KO and WT mice, and Figure 2D depicts weight fluctuations in female KO and WT mice.

Fifteen-week-old male KO and WT mice (*n* = 4) were dissected, and their internal organs were weighed. The mean weight of the liver and intestines of KO mice (2.03 ± 0.5 g and 3.5 ± 0.9 g, respectively) were greater than those of WT mice (1.2 ± 0.16 g and 2.0 ± 0.14 g, respectively) (Figure 2E,F), but no significant differences were observed in the mean weight of their spleen, kidney, heart, brain, bladder and stomach (Figure 2E). The KO mice had greatly increased fat tissue around the pelvis and kidneys, as is further described in Table 3. However, no statistically significant differences were observed in the level of liver triglycerides (52.0 ± 27.8 µmol/gr in KO mice compared to 40.3 ± 15.2 µmol/gr in and WT mice) (Figure 2F) and cholesterol (22.6 ± 1.15 µmol/gr in KO mice and 20.3 ± 2.6 µmol/gr in and WT mice) (Figure 2G).

Perirenal adipose tissue was different between KO and WT male mice, as described in Table 3 below. Following the injection of Actrapid (human insulin, Novo Nordisk) to 30-week-old mice, mice were sacrificed after 30 min and testicular fat, kidney fat and liver were assessed. Total fat was determined as the sum of the testicular and kidney fat, and the % fat/BW was determined as the sum of testicular and kidney fat/body weight * 100%.

### 3.3. Greater Insulin Sensitivity in Male KO Mice

Whole-body insulin sensitivity was measured by the relative blood glucose reductions in response to the insulin challenge test. Following insulin injections, glucose levels decreased by a rate of 1% per minute and at 40–45 min reached a maximum reduction of 70% in CD24 KO compared to baseline glucose levels, at which time they started to increase.

Male KO mice demonstrated 10–20% higher insulin sensitivity than male WT mice (Figure 3A). While no statistically significant differences were demonstrated in baseline glucose levels (data not shown), following insulin injection, the decrease in whole-blood glucose levels, was 20% and 10% greater in the KO males than in the WT males at 30 and 45 min, respectively (*p* < 0.05 for both). These observations were also gender-dependent, as no statistically significant differences were observed for the insulin challenge test between female KO and WT mice (Figure 3B).

### 3.4. Blood Glucose Response to IP Glucose Challenge in KO Mice

Blood glucose response was measured by the relative blood glucose increase in response to glucose stimulation. Baseline plasma insulin concentrations, drawn from 9–12 weeks old male mice, did not differ significantly between the two genotypes (Figure 4A) while at 20 min post-injection insulin levels were lower in the KO mice (Figure 4B).

### 3.5. High Expression of PPARγ in Male KO Mice

Since PPARγ has been associated with the regulation of insulin sensitivity, we compared the levels of PPARγ expression in perirenal fat depot tissue of male KO and WT mice. As shown in Figure 5A, in KO mice, PPARγ mRNA levels were 1.5 times higher than in WT mice. In addition, quantitative real-time PCR was carried out, confirming these results (Figure 5B). It is noteworthy that the visceral fat in the obese CD24 KO mice differed not only from that of controls, but also from classical forms of obesity models such as diet-induced obesity. Therefore, we also evaluated, by real-time PCR, the levels of Perilipin-1, Adiponectin and PPARα (Figure 5C). Adiponectin expression, which is typically low in the obese state [33] was similar to that in the lean control mice; PPARα expression was markedly lower, as was Perilipin-1 expression, despite larger fat mass and larger adipocyte size. Because Perilipin-1 is actively involved in lipolysis and PPARα in fatty acid oxidation, these findings are consistent with the observation of enhanced insulin sensitivity in the CD24 KO obesity model: insulin suppresses adipocyte lipolysis, such that fat cell lipolytic cascades are relatively inhibited, thus requiring lesser expression in proteins comprising the lipolysis machinery.

### 3.6. Differences in Adipocyte Size

Adipocyte increased cell size (hypertrophy) has been associated with obesity in humans [19,22]. Therefore, we compared adipocyte size between KO and WT genotypes. As shown in Figure 6, KO mice showed statistically significant increase in adipose cell size, as demonstrated through the white adipocyte tissue (WAT) cell area of 8258 ± 2359 µm^2^ in KO mice and 5471 ± 2030 µm^2^ in WT mice.

### 3.7. Differences in Enteric Bacterial Populations

High throughput sequencing of stool 16S rRNA genes was evaluated in the enteric microbial populations of male and female mice of either KO or WT genotype, that were fed either a normal or a high fat diet. Enteric bacterial populations were significantly different between young male KO and WT mice that were fed a normal diet by unweighted (R = 0.32, *p* < 0.01) β-diversity analysis. These differences became much more apparent when mice were kept on a high-fat diet by weighted (R = 0.43, *p* < 0.01) and un-weighed β-diversity analysis (R = 0.31, *p* < 0.01) (Figure 7). Specifically, in young KO males, many more bacterial strains were overrepresented when they were fed a high-fat diet than when they were fed a normal diet (24 strains vs. 4 strains respectively, each with an LDA score of 2.4 or more (data not shown). No significant differences were found in the other test groups (data not shown).

## 4. Discussion

This study provides new insight into the role of CD24 in insulin sensitivity and obesity. HSA/CD24 deficiency was found to be associated with increased body weight and visceral obesity only in male KO mice as compared to WT mice, despite similar food and water consumption in both groups. Only male KO mice demonstrated increased insulin sensitivity and increased expression of PPARγ.

The current model is interesting as it demonstrates the early onset of male obesity along with increased insulin sensitivity. This increased sensitivity might be mediated through the PPARγ pathway, which is known to be involved in fatty acid storage and the regulation of glucose metabolism. The activation of PPARγ in mature adipocytes induces a number of genes involved in the insulin-signaling cascade, thereby increasing insulin sensitivity [34]. In addition, the induction of adipogenesis, associated with the capability for fatty acid trapping, has been shown to contribute to the maintenance of systemic insulin sensitivity [35].

The higher expression of PPARγ in male KO mice, as compared to WT mice, suggests an association between CD24 and PPARγ, and indicates that CD24 may be a suppressor of PPARγ in male mice. This observation supports the documentation of increased PPARγ expression in CD24-negative cells of male mice compared to CD24-positive cells [35]. In that study, CD24-positive cells were identified as adipocyte progenitors, which, upon becoming further committed to the adipocyte lineage, lost CD24 expression and generating CD24-negative pre-adipocytes, the latter becoming discernible only after birth. The population shift of CD24-positive to CD24-negative cells from the prenatal to postnatal stages may explain the seeming discrepancy between studies, such as the one described above, regarding the insulin-producing capacity of CD24-negative cells [16,36]. Furthermore, the loss of CD24 expression appears to be a stage of the process by which mature adipocytes are formed from CD24-positive cells [37]. The increased insulin sensitivity in CD24-negative cells, demonstrated in the present study, suggests that the population shift from CD24-positive to CD24-negative cells may increase insulin sensitivity, as well as stimulate adipogenesis. Conversely, high expression of CD24-positive cells may be associated with insulin resistance and impaired adipogenesis. In this context, it is interesting to note that CD24 does not affect glucose uptake in differentiating adipocytes in vitro [37,38,39,40]. Considering the associations documented between CD24 and cancer [41], we suggest that further investigation of the role of CD24 in insulin sensitivity and weight gain may elucidate associations between cancer and both diabetes and obesity [26].

The present study shows that adipocytes in CD24 KO mice are hypertrophic. In contrast, previous research has demonstrated that the loss of CD24 has been associated with hypotrophy in mice, but that this phenotype is reversed into hypertrophy when fed a high-sucrose or a high-fat diet [14]. One explanation may be that the absence of CD24 has a different effect on different sources of adipocytes, as the latter study assessed interscapular, inguinal and epididymal white adipocyte tissue (WAT). Further investigation is needed to understand the mechanisms by which CD24 and adipocyte phenotypes are linked.

Explanations for gender-specific effects of PPARs may lie in differences in body fat, dietary habits and nutrient metabolism, hormonal activity, and in differential PPAR activity that may be related to other factors. Gender-specific differences in nutrient metabolism, such as a higher capacity for storing fat in adipose tissue and for oxidizing fatty acids in muscle in females, as demonstrated in rats [42], may be related to the gender-specific effects of PPAR activity on BMI and fat mass. Gender-specific expression of hormones, transcription factors, and genes may also affect PPAR activity [43]. In a study of sex differences in subcutaneous adipose tissue transcriptional regulation, 162 genes were found with robust sex-related expression differences. Interestingly, the genes that were found were enriched for binding motifs for adipo-genic transcription factors, including PPARγ [41,42,43,44]. Furthermore, in rats, sex hormones appear to affect the expression of the PPARγ2 subtype in adipose tissues [45]. Ciana et al. reported considerably lower PPAR transcriptional activity in female rats than in males; a high fat diet, gonad removal, and hormone replacement did not increase activity [46]. These and other findings indicate that, although gender is determined genetically, it can be considered an environmental factor that modifies both penetrance and expressivity of traits [47], and it therefore should be considered in any study of PPAR activity.

Investigations of the association of CD24 with obesity and weight gain in humans should consider the distribution of the single nucleotide polymorphisms (SNPs) of PPARγ, including gender-dependent associations in this regard. For example, female, but not male carriers of the PPARγ^C161T^ allele, were shown to have a higher mean body weight and waist circumference; male C161T allele carriers had lower insulin levels than male non-carriers [38]. Computational analysis identified five PPARγ variants expressed in cancer tissues and associated with insulin resistance and partial lipodystrophy, including C162S, R166W, Q286P, Q314P and P467L. Specifically, the effects of the C162S variant found in this study, may be similar to the effects of the C161T allele due to their immediate proximity in the protein structure [48]. In addition, a number of studies have associated the PPARγ^P12A^ allele with reduced BMI [49,50,51], whereas others have associated it with increased BMI [52,53,54,55,56,57]. Most of these studies did not stratify their analysis by gender. More conclusively, the P12A allele has been associated with lower fasting insulin levels [58], increased insulin sensitivity [49,58,59,60,61,62] and reduced risk of T2D [63,64,65,66,67,68,69]. In addition, we have previously associated the C248T SNP with statistically significant higher blood levels of total cholesterol and LDL-C (unpublished data), suggesting that CD24 may also play a role in dyslipidemia, possibly through regulation of PPARγ expression.

PPARγ signaling has been linked with gut microbiota in several studies. For example, PPARγ signaling is involved in homeostasis of gut microbiota as it controls and limits the expansion of potentially pathogenic dysbiotic bacteria [70]. Furthermore, a recent study showed that PPARγ is strongly affected by the metabolites secreted by commensal bacteria, at least several of them through a change in PPARγ’s phosphorylation status [71]. Interestingly, in a study of the oral epithelial immuno-transcriptome response by a multispecies biofilm, CD24 expression was elevated under bacterial challenge. It would be of interest to explore whether CD24 expression is regulated by the gut microbiome as well, and if PPARγ-mediated microbiome homeostasis plays a role in this purported CD24 regulation.

It is possible that the larger liver mass in KO mice, compared to WT mice, may explain, at least in part, the greater insulin sensitivity demonstrated in this study. Finally, the possibility of a role for CD24 in insulin sensitivity and obesity, as suggested in this study, may contribute to an improved understanding of the increasingly growing association between cancer and both diabetes and obesity.

In this study, we aimed to assess gender-dependent changes in CD24 KO and its association with PPARγ expression. Insulin sensitivity, early obesity, and PPARγ expression were assessed in CD24 knockout (KO) mice and compared to wild-type (WT) mice. Our results demonstrate gender-specific differences in insulin sensitivity. CD24 KO male mice displayed, at an early age, greater insulin sensitivity and glucose uptake, suggesting a gender-dependent role of CD24 in insulin-sensitive obesity. We conclude that CD24 may negatively regulate PPARγ expression in male mice. Furthermore, the association between the CD24 and insulin sensitivity suggests a possible mechanism for diabetes as a cancer risk factor. Finally, CD24 KO male mice may serve as a model of obesity and insulin hyper-sensitivity.

## Figures and Tables

**Figure 1 jpm-11-00050-f001:**
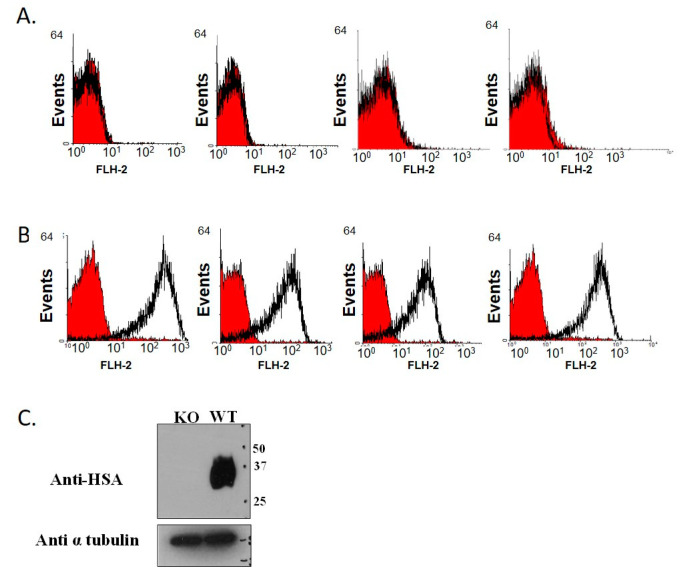
Genotype verification by FACS analysis. Heparinized blood samples from KO (**A**) and WT (**B**) mice were collected and analyzed for CD24 expression by FACS analysis. The red curves represent the negative control (secondary antibody only), and the black curves represent the binding of M1/69 anti-CD24 antibody. (**C**) Blood samples were taken from KO and WT mice and peripheral blood leukocytes were isolated. Protein extracts (20 µg) were subjected to SDS-PAGE (odium dodecyl sulphate–polyacrylamide gel electrophoresis) and Western blotting using anti-CD24 M1/69 monoclonal antibodies. The membrane was re-probed with anti-tubulin to confirm uniform loading of the samples.

**Figure 2 jpm-11-00050-f002:**
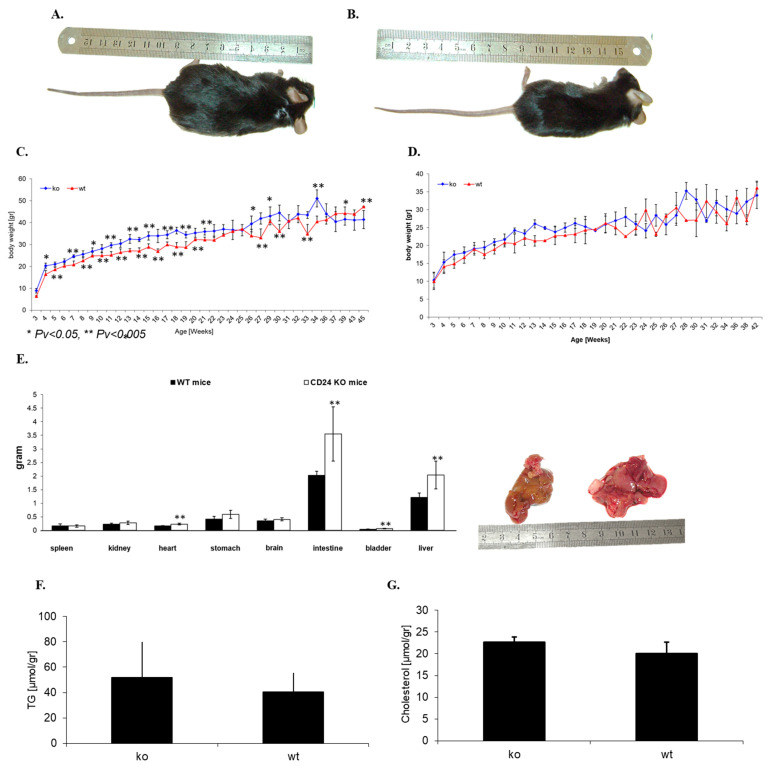
Whole body and organ weights. Representative pictures of male, 15-week-old KO (**A**) and WT (**B**) mice demonstrate the differences in body weight. Both male (**C**) and female (**D**) KO and WT mice were monitored, and their body weights were recorded from birth. Four mice, 15 weeks old, were dissected and their internal organs weighed separately (**E**). Liver triglycerides (**F**) and cholesterol (**G**) were measured in KO and WT mice.

**Figure 3 jpm-11-00050-f003:**
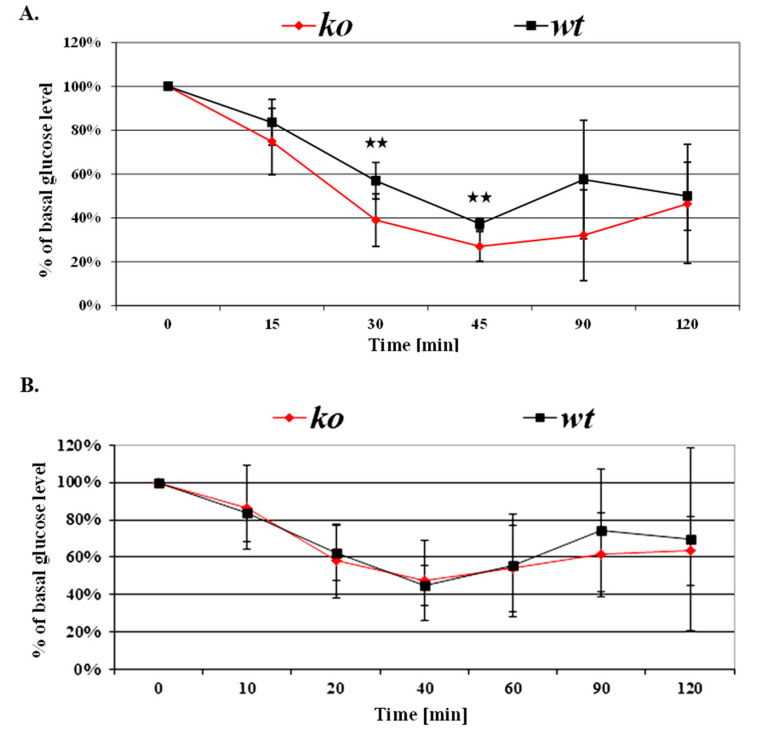
CD24 deficiency and insulin sensitivity. Insulin sensitivity of KO and WT males and females, aged 9–12 weeks, is presented in (**A**,**B**), respectively. Following intraperitoneal insulin injection (0.5 unit/kg body weight) of fasted mice, whole-blood glucose was measured at the indicated time points using a glucometer. Values are expressed as a percentage, relative to baseline glucose levels. Asterisks indicate statistically significant differences (*p* < 0.05) between genotypes at each time point. Each value represents the average of at least three independent measurements.

**Figure 4 jpm-11-00050-f004:**
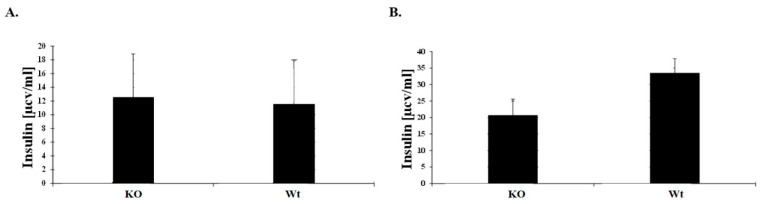
CD24 deficiency and glucose uptake in 9–12 weeks old mice. After fasting, mice were intraperitoneally injected with glucose (1 g/kg body weight). (**A**) Baseline insulin levels were measured from mice serum by ELISA assay. (**B**) Insulin levels were measured 20 min after glucose injection.

**Figure 5 jpm-11-00050-f005:**
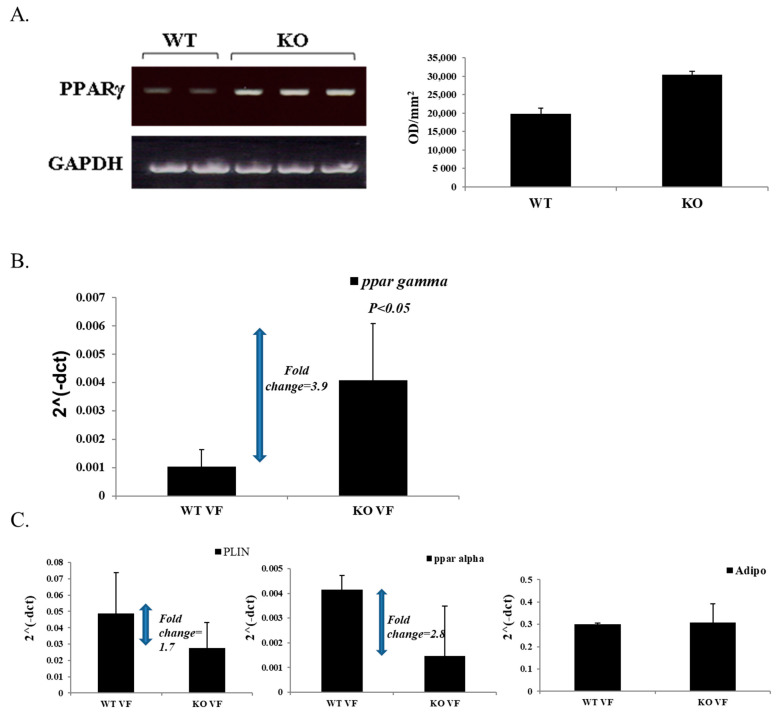
PPARγ expression in KO and WT male mice. RNA extraction from kidney adipose tissue from KO and WT mice was prepared and cDNA synthesis was performed. (**A**). A fragment of the PPARγ gene was amplified by PCR reaction and uniformity of the samples was confirmed by the murine GAPDH housekeeping gene. PPARγ expression levels in samples, shown on the right, were determined by densitometry (TINA 2.0). (**B**). Quantitative real-time PCR was performed to confirm the results above. (**C**). Quantitative real-time PCR was performed to determine the levels of perilipin 1, PPARα, and adiponectin in visceral fat.

**Figure 6 jpm-11-00050-f006:**
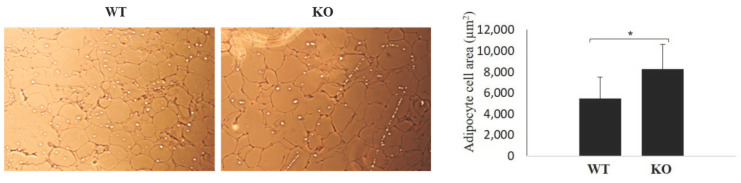
Adipocytes in KO and WT mice. Adipocytes were significantly larger in KO mice than in WT mice. Brightfield imaging was done with a Nikon microscope at ×10 magnification. * *p* < 3.51 × 10^9^.

**Figure 7 jpm-11-00050-f007:**
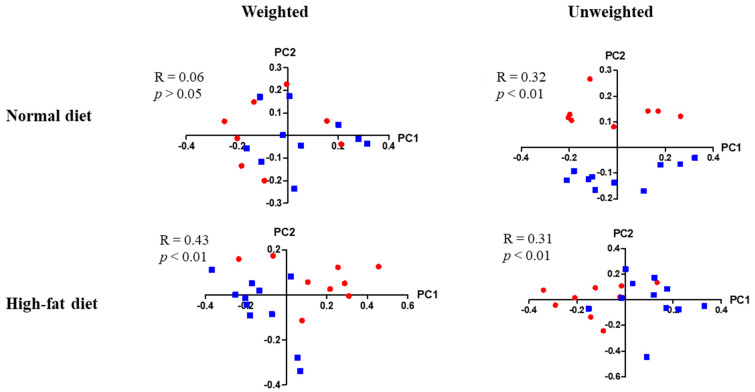
Enteric bacterial populations in young male KO and WT mice. Red circles indicate KO mice, whereas blue squares indicate WT mice. For mice fed a normal diet, *n* = 8 KO mice, and *n* = 10 WT mice. For mice fed a high-fat diet, *n* = 9 KO mice, and *n* = 11 WT mice. Correlation coefficients and *p*-values are shown in the figure.

**Table 1 jpm-11-00050-t001:** Rodent Diet with 60% kcal Fat Formulation.

**Class Description**	**Ingredients**	**Gram**
Protein	Casein, Lactic, 30 Mesh	200
Protein	Cystine, L	3
Carbohydrate	Lodex 10	125
Carbohydrate	Sucrose, Fine Granulated	72.8
Fiber	Solka Floc, FCC200	50
Fat	Lard	245
Fat	Soybean Oil, USP	25
Mineral	S10026B	50
Vitamin	Choline Bitartrate	2
Vitamin	V10001C	1
Dye	Dye, Blue FD&C #1, Alum. Lake 35–42%	0.05
**Caloric Information**		
Protein	20% Kcal	
Fat	60% Kcal	
Carbohydrate	20% Kcal	
Energy density	5.21 Kcal/g	

**Table 2 jpm-11-00050-t002:** Oligonucleotides sequence.

Name	Sequence (5′ → 3′)	PCR Reaction
Neo-cassette-Forward	TTGAACAAGATGGATTGCACGCA	95 °C—5 min		
Neo-cassette-Reverse	TGATCGACAAGACCGGCTTCC	95 °C—1 min65 °C—1 min72 °C—30 s	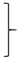	x34 cycles
72 °C—10 min		
Exo1-Forward	TAGCAGATCTCCACTTCCG	95 °C—5 min		
Exo1-Reverse	GTAGGAGCAGTGCCAGAAGC	95 °C—1 min60 °C—1 min72 °C—15 s	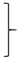	x34 cycles
72 °C—10 min		

**Table 3 jpm-11-00050-t003:** Fat coefficient evaluation.

	BW ± SE [gr]	Testicular Fat ± SE [gr]	Kidney Fat ± SE [gr]	Total Fat ± SE [gr]	%Total Fat/BW ± SE [%]
HSA^−/−^ mice (*n* = 6)	39.71 ± 2.53	2.07 ± 0.26	0.78 ± 0.14	2.86 ± 0.12	7.20 ± 0.19
HSA^+/+^ mice (*n* = 6)	30.31 ± 1.77	0.9 ± 0.11	0.26 ± 0.11	1.16 ± 0.11	3.85 ± 0.38

## Data Availability

Not relevant.

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
