# Peer review of "High Expression Level of PPARγ in CD24 Knockout Mice and Gender-Specific Metabolic Changes: A Model of Insulin-Sensitive Obesity"

_jpm, 2021, doi:10.3390/jpm11010050_

Round 1

Reviewer 1 Report

In this manuscript, Shapira and co-workers examined the metabolic phenotype of CD24 knockout mice focusing on PPARy expression, with particular regard to possible gender differences. By measuring body/tissue weights, insulin sensitivity and fat content they proposed CD24 knockout mice as a mouse model of "insulin-sensitive obesity".

The overall quality of presented data is low, basic informations are missing and many sections are skimpy.

In the abstract section basic informations (background, aim, experimental approach and conclusions) are missing or written in a alogical way.

The sentences on L.48-50 and L.55 are incomprehensible.

Fig 1C-D are incomprehensible. BW should be recorded 1 time per week and a single graph should be provided with four groups WT and KO male and female.

In the introduction section very few information on CD24 and the experimental model are present.

All figures must be improved since the quality is very low and basic informations are missing (e.g scale bar in FIG.6, y-axis legend in FIG.1E)

The material and methods section must be rewritten including all the information needed to understand how the experiment were performed. Informations about reagents and instruments are completely missing. Statistical analysis for all the presented data must be included.

No informations are provided about the CD24 knockout mouse model:e.g is a total knockout? It is unclear why authors used FACS for genotyping animals.

A semi-quantitative PCR for PPARy cannot be accepted in the 2020!

It is unclear way authors focused on perirenal adipose tissue. This choice should be at least discussed and justified. The increased fat mass must be assessed first of all by providing a fat coefficient (% FAT/BW) for each examined adipose tissue (visceral, subcoutaneous)

It is difficult to believe that food and water consumption "were closely monitored from birth" as stated by authors on L.87-88

References must be revised and standardized (e.g L.64 and L.181)

No informations about the HFD were included (e.g type, composition, administration time)

Thus, this manuscript does not meet the standard for a publication in Journal of Personalized Medicine.

Reviewer 2 Report

I appreciate the opportunity to review this interesting paper on CD24 knockout mice.

The authors studied gender-dependent changes in CD24 KO mice and its association with PPARγ expression.

They found that male, but not female KO mice showed increased insulin sensitivity and glucose uptake compared to WT mice, as well as higher PPARγ expression. They also found that adipocytes were larger in male KO mice than in WT mice.

I commend the authors for several strengths of their work, including addressing an interesting and timely problem and the good conduct of experiments.

The subject is in the range of the journal, and the manuscript is well written, and data are appropriately presented.

Considering these strengths, though, as I read the manuscript, I found some areas in which I would have appreciated greater clarity.

  • The introduction lacks information on CD 24 and its role in physiological and pathological processes.
  • The introduction should not discuss the results or conclusions. On the other hand, there is no clearly defined research goal and correctly formulated research questions. It would appear from the abstract that the authors' aim would be to assess gender differences, but in the introduction, they do not refer to it at all. Nor do they explain why it would be important to study these gender differences in mice.
  • What are the compositions of the standard and the high-fat diet (HFD)? The composition of the diets should be provided. It should be indicated how much fat is in this diet. Does the standard diet deliver the same amount of calories (and has a different carbohydrate to fat ratio) as the high-fat diet? What is the source of the fat? What is the concentration of cholesterol in the diet? In case the authors have fed a chow diet as a standard diet and compare their diet with a defined high-fat diet, the detected differences may arise from the “undefined” compounds present in the chow diet.
  • It is essential to show the list of food intake, body weight, and body composition (e.g., body fat content) of the animals. A difference in energy intake (or also in food intake per se) may be the reason for the observed outcome in the present study.
  • A flow chart showing the experimental procedure should be included as it makes the experimental setting more visible to the reader.
  • Obesity is characterized by specific alterations in the composition and function of the gut microbiome. The gastrointestinal microbiota can influence both sides of the energy balance equation; namely, as a factor influencing energy utilization from the diet and as a factor that influences host genes that regulate energy expenditure and storage. Recent research has focused on the influence of HFD consumption on gut microbial composition. For example, Carmody et al. [1], used over 200 strains of mice to determine whether variations in gut microbiota are primarily driven by host genetics or by dietary factors. Their findings indicate that a high-fat reproducibly altered the gut microbiota despite differences in host genotype. For example, it has been reported that HFD promotes a decrease in Bacteroidetes and an increase in both Firmicutes and The authors should describe the changes observed in the composition of the microbiota more precisely and refer their observations to the literature.
  • An interesting observation is the finding of obesity and, at the same time, increased insulin sensitivity in the tested mice. Obesity with insulin sensitivity has already been reported, but in such cases, no adipocyte hypertrophy has been observed. Neither has been observed in such cases, the pathological profile of adipokines, which is a good predictor of insulin resistance. I understand that adipokines were not tested in the current research? Taking into account the certain controversy of these observations, this issue should be given much more space in the discussion.
  • I understand that adipokines were not tested in the current research?
  • Carmody, R.N.; Gerber, G.K.; Luevano Jr, J.M.; Gatti, D.M.; Somes, L.; Svenson, K.L.; Turnbaugh, P.J. Diet dominates host genotype in shaping the murine gut microbiota. Cell host & microbe 2015, 17, 72-84.

Reviewer 3 Report

In their study, Shapira et al. presented a very interesting topic of the role of CD24 in the pathogenesis of insulin resistance development. 

The introduction and methods sections are written well, with a clear indication of the aim of the study and the full methodology used.

The results section presents in a clear way the outcomes of the study. However, I have a really minor, editorial comment, to work a little bit on the style of your figures - in some of them the unit on the y-axis is missing and there are some spelling errors in x-axis labels. Moreover, I would consider adding the first paragraph of the Results section (on how the CD24 mice were established) into supplementary materials.

In the discussion, authors extensively comment on their findings and compare to the current state of the knowledge. However, in my opinion, at the end of the paragraph, you should add a short summary of your study as a conclusions section.

Round 2

Reviewer 1 Report

The authors have adequately addressed my previous comments and questions. No further concerns.

Author Response

Thank you for your comments and recommendation for publishing our article

Reviewer 2 Report

The authors have addressed satisfactorily most of the points raised by the reviewer.

The authors have addressed satisfactorily most of the points raised by the reviewer. Than you.

Author Response

(The authors gave the same response as above.)
